# The Actigraphy-Based Identification of Premorbid Latent Liability of Schizophrenia and Bipolar Disorder

**DOI:** 10.3390/s23020958

**Published:** 2023-01-14

**Authors:** Ádám Nagy, József Dombi, Martin Patrik Fülep, Emese Rudics, Emőke Adrienn Hompoth, Zoltán Szabó, András Dér, András Búzás, Zsolt János Viharos, Anh Tuan Hoang, Bálint Maczák, Gergely Vadai, Zoltán Gingl, Szandra László, Vilmos Bilicki, István Szendi

**Affiliations:** 1Department of Software Engineering, University of Szeged, 13 Dugonics Square, 6720 Szeged, Hungary; 2Department of Computer Algorithms and Artificial Intelligence, University of Szeged, 2 Árpád Square, 6720 Szeged, Hungary; 3Doctoral School of Interdisciplinary Medicine, Department of Medical Genetics, University of Szeged, 4 Somogyi Béla Street, 6720 Szeged, Hungary; 4ELKH Biological Research Centre, Institute of Biophysics, 62 Temesvári Boulevard, 6726 Szeged, Hungary; 5Institute for Computer Science and Control, Center of Excellence in Production Informatics and Control, Eötvös Lóránd Research Network (ELKH), Center of Excellence of the Hungarian Academy of Sciences (MTA), 13-17 Kende Street, 1111 Budapest, Hungary; 6Faculty of Economics and Business, John von Neumann University, 10 Izsáki Street, 6000 Kecskemét, Hungary; 7Department of Technical Informatics, University of Szeged, 2 Árpád Square, 6720 Szeged, Hungary; 8Department of Psychiatry, Kiskunhalas Semmelweis Hospital, 1 Dr. Monszpart László Street, 6400 Kiskunhalas, Hungary

**Keywords:** machine learning, actigraphy, bipolar disorder, schizophrenia, linear regression, light gradient boost, random forest

## Abstract

(1) Background and Goal: Several studies have investigated the association of sleep, diurnal patterns, and circadian rhythms with the presence and with the risk states of mental illnesses such as schizophrenia and bipolar disorder. The goal of our study was to examine actigraphic measures to identify features that can be extracted from them so that a machine learning model can detect premorbid latent liabilities for schizotypy and bipolarity. (2) Methods: Our team developed a small wrist-worn measurement device that collects and identifies actigraphic data based on an accelerometer. The sensors were used by carefully selected healthy participants who were divided into three groups: Control Group (C), Cyclothymia Factor Group (CFG), and Positive Schizotypy Factor Group (PSF). From the data they collected, our team performed data cleaning operations and then used the extracted metrics to generate the feature combinations deemed most effective, along with three machine learning algorithms for categorization. (3) Results: By conducting the training, we were able to identify a set of mildly correlated traits and their order of importance based on the Shapley value that had the greatest impact on the detection of bipolarity and schizotypy according to the logistic regression, Light Gradient Boost, and Random Forest algorithms. (4) Conclusions: These results were successfully compared to the results of other researchers; we had a similar differentiation in features used by others, and successfully developed new ones that might be a good complement for further research. In the future, identifying these traits may help us identify people at risk from mental disorders early in a cost-effective, automated way.

## 1. Background

### 1.1. Introduction

Schizophrenia and bipolar disorder are mental disorders with overlapping susceptibility genes, structural and functional brain abnormalities, symptoms, and treatment response [1]. In the last couple of decades, there has been a significant improvement in treating schizophrenia and bipolar affective disorder. In the early 2000s, an approach was applied that helped to identify the first psychotic episode in patients much earlier, so their treatment could be started sooner. In the first couple of years, their results seemed promising, but overall, it did not prevent the chronic progression in many of the patients; hence, from the point of view of the outcome, the improvement was only moderate [2]. Therefore, the focus was shifted to even earlier periods before the first psychotic episode [3,4]. According to earlier studies, this prodromal phase occurs in 73–98.4% of the patients and can last on average up to 5–6 years [5,6]. During this time, the chance opens up for prevention, which is becoming the focus in psychiatry as well.

To forecast the onset of a psychosis or a psychotic disorder, we have two methods. One approach is the ultra high risk (UHR) criterion that helps to define an imminent risk of psychosis onset within one year [7,8]; the other one is the basic symptoms (BS) method, that is more specific for indicating the development of schizophrenia [9]. However, these methods can only be used among those patients who already had some mental complaints, hence asked for help but cannot be utilized for screening healthy populations. Besides this, there is still a reversed relationship between the specificity and sensitivity of the predictive criteria, which lead to clinical and ethical dilemmas for doctors [10]. The methods of repeated assessments of the help-seeking individuals’ clinical states with the designation of syndrome stages, and multivariate analyses of empirically derived markers have proven to be useful tools for finding a balance between the sensitivity and specificity [11].

There have been some studies conducted on the sleep and daily functions of people with the schizophrenia-bipolar spectrum disorders. Reeve et al. [12] found that 80% of their subjects with early, non-affective psychosis had some kind of severe (either chronic or frequent) sleep disorder, for example, 50% had insomnia. Those patients who had problems with their sleep were suffering from more severe psychotic symptoms. Mulligan et al. [13] also found a connection between sleep problems (fragmentation, reduced efficiency) and psychotic symptoms on the following day. Meyer et al. [14] studied patients with remitted schizophrenia and bipolar disorder, and they also found that sleep problems (such as fragmentation) was something patients had to deal with more compared to healthy controls.

When daily activity was in focus, it was found to be reduced in patients with remitted schizophrenia and bipolar disorder [14,15]. Moreover, euthymic bipolar patients displayed activity patterns that could be described as less organized or predictable, especially after a life event or being in a novel environment. Euthymic bipolar patients could be differentiated from other groups in the variability and complexity of activity patterns and they had a relative lack of habituation [15].

Researchers also focused on people who did not yet have a differentiated disorder but were at risk (e.g., showing some symptoms, having a diagnosed close relative, etc.) of developing one. Hennig et al. [16] found that using multilevel regression models, shorter sleep time and more dreaming could predict paranoia on the following morning in patients who had some psychotic symptoms, but not full-blown psychosis. Lunsford-Avery et al. [17] found that ultra-high risk (UHR) people moved more during sleep, woke up more often after sleep onset, and slept less efficiently compared with healthy controls. Some of the sleep problems could be used to predict psychotic symptoms in the 12-month follow-up.

Castro et al. [18] found that circadian rest–activity rhythms (and likely sleep patterns) might be altered in UHR (to schizophrenia and borderline disorder) individuals compared to healthy controls, as in the UHR group worse sleep quality, higher total sleep time, more fragmented sleep rhythms, longer nap duration, and shorter activity periods were found. According to the authors, it is probable that the alterations in circadian rhythms of activity and rest could be prodromal signs and might imply the brain changes which are common in these disorders.

Ritter et al. [19] carried out a longitudinal study in which they utilized data from a large cohort (N = 1943) free of mental disorder used as the baseline. They found that poor sleep quality increased the risk of developing bipolar disorder, trouble falling asleep, and early morning awakening being predictive factors. Meyer and Maier [20] placed their subjects in the bipolar risk group if they had a hypomanic personality; the unipolar risk group consisted of subjects with more rigid personality traits; and the control group had subjects with none of these traits. The bipolar risk group had less regularity with daily activities according to their diaries and more variability in their sleep pattern compared to the control group.

The aim of our study is to help improve the chance of sensitivity and specificity with instrumental procedures in order to recognize the path of pathological schizotypal and bipolar disease development early and to treat people at risk as early as possible. In this study, our goal was to screen a healthy population, to detect individuals at low risk with bipolar and schizotypal vulnerability based on personality traits and to attempt to differentiate them via certain instrumental methods. To achieve this, we used a methodology called actigraphy. Actigraphy [21] is a diagnostic methodology that tracks overall activity during sleep–wake cycles by measuring the triaxal acceleration of limb movements using an accelerometer sensor attached to the limb. It can be used to calculate the patient’s resting/active period, which can then be used to monitor sleep quality and validate sleep–wake cycles. In the first step, using screening tests sensitive to bipolar affective risk and schizotypy detection, three homogeneous subgroups were formed among those with traits predisposing to one or the other disease, as well as those without such traits. In the second step, we investigated whether the results of the self-administered screening methods based on the testing of subjective experience changes could be linked to the empirical indices of circadian activity and sleep regulation that can be measured by actigraphy. Then, a confrontational analysis of actigraphic parameters was performed by comparing the groups showing a latent tendency towards the development path of the two diseases in order to determine specific early markers.

### 1.2. State of Art

Recent actigraphy studies indicate the disturbance of sleep and circadian rhythm both in schizophrenia and bipolar disorder [14,22,23,24,25,26]. In terms of circadian rhythm parameters, schizophrenia and bipolar patients have lower motor activity compared with healthy controls [14,22,26]. The manic phase of bipolar disorder is associated with more disorganized and complex activity patterns, especially in the morning, and with phase advance [24]. The depressive phase is characterized by higher minute-to-minute variability in activity pattern, and phase delay [23,24]. In the case of schizophrenia, more severe positive symptoms were associated with less structured movement patterns, and more severe negative symptoms were associated with lower motor activity [26].

According to Berle et al. [27], circadian rhythm parameters interdaily stability was higher and intradaily variability was lower in the schizophrenia patient group than in the healthy control group, suggesting the motor activity pattern of schizophrenia patients is more monotonous. Jones, Harc, and Evershed (2005) found that intradaily variability was higher and interdaily stability was lower among bipolar patients compared to healthy controls [28]. However, other studies did not find any significant mean difference in interdaily stability, intradaily variability, or in other circadian rhythm parameters such as the relative amplitude and acrophase among schizophrenia, bipolar, and healthy control groups [14].

Sleep disturbance manifests itself in longer total sleep time, a longer time in bed, longer sleep latency and a longer time being awake after sleep onset both in schizophrenia and bipolar disorder compared with healthy controls, and reduced sleep efficiency in bipolar disorder compared with healthy controls [14,22]. Schizophrenia and bipolar patients compared with healthy controls need more time to fall asleep, spend more time in bed, and have more fragmented sleep [14,26]. More disrupted sleep–wake patterns are associated with more severe positive symptoms of schizophrenia. More severe negative symptoms are associated with longer sleep time, more common awakenings during the night, and sleepiness during the day. The sleep disturbance in schizophrenia and bipolar disorder causes further increase in the symptoms of the disorders [26].

### 1.3. Data Collection

The following questionnaires were used in the study for screening: Temperament Evaluation of Memphis, Pisa, Paris, and San Diego Autoquestionnaire (TEMPS-A) [29] is a self-report questionnaire assessing the five affective temperaments: depressed, cyclothymic, irritable, hyperthymic, and anxious. The questionnaire consists of 110 items for women and 109 for men, with all questions requiring a “yes” or “no” response. The 43-item Hungarian version of the shortened Oxford-Liverpool Inventory of Feelings and Experiences (O-LIFE) is used to assess schizotypal personality traits [30]. The items of the questionnaire can be answered with “yes” or “no”. The short version of the questionnaire consists of four scales: Unusual Experiences, Cognitive Disorganization, Introverted Anhedonia, and Impulsive Nonconformity. The Unusual Experiences scale measures the positive trait of schizotypy with items related to perceptual abnormalities, magical thinking, and hallucinations. The Cognitive Disorganization scale includes questions about attention and concentration difficulties, decision-making problems, and social anxiety. The Introverted Anhedonia scale assesses negative schizotypy or schizoid temperament, which can be described as the reduced form of negative symptoms in schizophrenia. The Impulsive Nonconformity scale contains items measuring lack of self-control [31]. The Peters et al. Delusions Inventory (PDI) [32] measures delusions that occur in the normal population. The questionnaire contains 40 questions on different types of delusions (delusions of control, expansive delusions, etc.). Each item of the PDI is rated on a five-point Likert scale under three different aspects: distress (from “not at all distressing” to “very distressing”), preoccupation (from “hardly ever think about it” to “think about it all the time”), and conviction (from “don’t believe it’s true” to “believe it’s absolutely true”).

The Mood Disorder Questionnaire is a self-report questionnaire used to screen for bipolar spectrum disorder [33]. The first item of the MDQ consists of 13 yes/no questions to screen for manic or hypomanic symptoms of bipolar disorder. The questionnaire contains two additional items related to whether the respondent experienced multiple symptoms of the above 13 manic or hypomanic symptoms at the same time and the extent to which these symptoms caused problems, which were rated on a 4-point Likert scale (from “no problem” to “serious problem”).

The Clinician Version (SCID-CV) of the Structured Clinical Interview for DSM-5 (SCID-5) was used to diagnose current and lifetime mental disorders [34]. The SCID-5-CV captures mental disorders such as mood disorders, psychotic disorders, substance use disorders, anxiety disorders, obsessive-compulsive and related disorders, eating disorders, somatic symptom disorders, sleep disorders, externalizing disorders, trauma-related disorders, and stressor-related disorders. The Structured Clinical Interview for DSM-5 Personality Disorders (SCID-5-PD) assesses the personality disorders listed in the DSM-5.

Demographic data, family and personal medical history, medications currently being taken, previously diagnosed and treated mental disorders, and family history of symptoms of psychosis or affective spectrum disorder were also recorded.

The tools used in research but not analyzed here include: Agency Attribution Task (developed by our research team); Intentional Binding Task [35]; Examination of Anomalous Self-Experiences (EASE) [36]; Discrimination of time intervals (DIS), and production of a single time interval (STT) [37]; the Temperament and Character Inventory-Revised (TCI-R) [38]; the Behavioral Inhibition and Activation System Scales (BIS/BAS Scales) [39]; Morningness–Eveningness Questionnaire [40]; Leuven Affect and Pleasure Scale (LAPS) [41]; the THINC-Integrated Tool (THINC-it) Screening Assessment for Cognitive Dysfunction [42]; Raven’s progressive matrices [43]; electroencephalogram (EEG); eye-tracker, assessment of allostatic load (heart rate, blood pressure, cortisol levels, C-reactive protein (CRP), interleukin 6 (IL-6), interleukin 12 (IL-12), total cholesterol; triglyceride levels).

### 1.4. Participant Selection

Study participants were recruited online via social media platforms and the University of Szeged mailing list. First, participants read general information about the study in a prompt email or online advertisement. They then clicked on a Web link included in the advertisement or email to access the study consent form. After agreeing to participate in the study, they were able to complete a screening package of questionnaires.

The subjects were healthy university students, who could not be diagnosed with any current mental disorder, from the first two years of the different faculties of the University of Szeged. In the first step, the intention was to form three homogeneous subgroups in the groups of subjects with low risk traits for schizophrenic-bipolar spectrum disorders. The test package consisted of 4 self-completion tests: the Peters et al. Delusions Inventory (PDI-21) [32] to measure delusions, including distress, preoccupation, and conviction; the Hungarian version [44] of the TEMPS-A (the Temperament Evaluation of Memphis, Pisa, Paris and San Diego—autoquestionnaire version) [29] questionnaire’s Cyclothymic Temperament scale; the Hungarian version [30] of the O-LIFE (the Oxford-Liverpool Inventory of Feelings and Experiences) questionnaire [45] measuring schizotypy; and the Hungarian version (unpublished) of the MDQ (the Mood Disorder Questionnaire) [33] assessing possible mood disorders. This primary survey was conducted electronically after recruitment through social media channels commonly used by university students.

Inclusion criteria of the Cyclothymia Factor Group (CTF, presumably with a latent tendency toward bipolarity): age 18–25 years, written informed consent, TEMPS-A Cyclothymia Score > median. Exclusion criteria: current presence of a primary or secondary mental disorder (SCID-1/2 DSM-5), Abbreviated O-LIFE Unusual Experiences score > median and PDI total score > median; MDQ scale minimum of 7 points for the first question, “yes” for the second question, and “at least a moderate” or “severe problem” for the third question; current abuse of psychoactive substances; a history of head trauma with permanent loss of consciousness; any physical illness known to affect brain structure; any unstable medical condition that may significantly impair neurocognitive function.

Inclusion criteria of the Positive Schizotypy Factor group (PSF, presumed to have a latent tendency toward schizotypy): age 18–25 years; written informed consent; abbreviated O-LIFE Unusual Experiences score > median; PDI total score > median. Exclusion criteria: current presence of a primary or secondary mental disorder (SCID-1/2 DSM-5); MDQ scale score of at least 7 for the first question, “yes” for the second question, and “at least a moderate” or “severe problem” for the third question; TEMPS-A Cyclothymia Score > median; current abuse of psychoactive substances; a history of head trauma with permanent loss of consciousness; any physical illness known to affect brain structure; any unstable medical condition that may significantly impair neurocognitive function.

Inclusion criteria for Control Group (C): age 18–25 years; written informed consent; abbreviated O-LIFE Unusual Experiences Score < median; PDI total score < median; TEMPS-A Cyclothymia Score < median. Exclusion criteria: current presence of a primary or secondary mental disorder (SCID-1/2 DSM-5); MDQ scale score of at least 7 points for the first question, “yes” for the second question, and “at least a moderate” or “severe problem” for the third question; current abuse of psychoactive substances; a history of head trauma with permanent loss of consciousness; any physical illness known to affect brain structure; any unstable disease state that may significantly impair neurocognitive function.

Based on the online questionnaire, we screened N = 710 students of the University of Szeged. Among the respondents, we selected N = 182 individuals based on the inclusion criteria. Based on the exclusion criteria, N = 87 students were eligible to form the three study groups. However, during the subsequent face-to-face medically structured diagnostic interview, we still had to exclude N = 2 individuals due to acute mental disorder confirmed on the basis of SCID.

Based on TEMPS-A, O-LIFE, and PDI values, participants were divided into three groups:A risk for bipolarity (hereafter, CTF group, Cyclothymia Factor Group) if their TEMPS-A Cyclothymia score was greater than 11, but their PDI-21 total score was less than 11 and their O-LIFE total score was less than 6;A schizotypy risk group (hereafter, PSF group, Positive Schizotypy Factor) included those with a PDI-21 total score greater than 10 and an O-LIFE total score greater than 5, but a TEMPS-A Cyclothymia Score less than 12;The Control Group included individuals who had relatively low scores on all three questionnaires, with a total score of less than 12 for TEMPS-A Cyclothymia, less than 11 for PDI-21, and less than 6 for O-LIFE.

To ensure comparability, the following groups were included in the study: CTF Group N = 25 students; PSF Group N = 26 students; Group C (control group) N = 29 students. We managed to obtain analyzable actigraphy data from 69 participants (33 male, 36 female): CTF Group (cyclothymia factor group, presumably showing a latent tendency to bipolarity) NCTF = 22, 11 male and 11 female, mean age MCTF = 25.28, standard deviation of age SDCTF = 1.78; PSF Group (positive schizotypy factor group, presumably showing a latent tendency to schizotypy) NPSF = 22, 11 male and 11 female, mean age MPSF = 26.20, standard deviation of age SDPSF = 2.06.; Group C (control group) NCONTROL = 25, 11 males and 14 females, mean age MCONTROL = 25.42, standard deviation of age SDCONTROL = 1.90. Eighty percent of the subject sample took no medications, 14% took birth control pills, 3% took beta blockers, and 3% took antihistamines.

The study was conducted as part of a study entitled “An examination of neurobiological, cognitive and neurophenomenological aspects of the susceptibilities to mood swings or unusual experiences of healthy volunteer students”, and it was approved by the Human Investigation Review Board of the University of Szeged, Albert Szent-Györgyi Clinical Centre, Hungary (No. 267/2018-SZTE) according to its recommendations. All subjects gave written informed consent in accordance with the Declaration of Helsinki and they were informed of their right to withdraw from the study at any time without providing any explanation. The selected participants received an expense allowance of HUF 15,000 for participation in the entire study, which was obtained through a grant application.

## 2. Method

### 2.1. Actigraphy Instruments

In a previous paper [46], members of our research team presented a novel actigraphy sensor used to measure and collect the required data. The device was housed in a 3D-printed capsule measuring 41 mm × 16 mm × 11.3 mm (LWH) and weighing only 5.94 g, which contained a microcontroller (C8051F410), a 3-axis MEMS accelerometer (LIS3DH), a flash memory chip with a capacity of 1 GB, and a real-time quartz clock with an accuracy tolerance of +/−20 ppm. It was necessary to develop our own device because, although there was already a number of actigraphy sensors available on the market, their price was very high at the beginning of our study, and we needed to provide sensors for dozens of patients. Nevertheless, our methodology and results are compatible with the actigraphic sensors available on the market.

The sampling rate and other options of the sensors are configurable, with a minimum sampling rate of 1 sample/second and a maximum sampling rate of 100 samples/second, and the flash memory is capable of storing measurements of entire weeks. This custom-built actigraphy instrument was able to acquire all the signals provided by commercial actigraphs, and it also provided the research team with full access to the different configurations and processing flow. This instrument was used to perform 84 measurements over a 4-month period in 2019.

### 2.2. Data Cleaning and Preparation

The first step in data preparation was to keep only those measurements where the instrument was able to gather data for at least 2 days without any discharges or problems (measurements lasted 14 days on average). In this way, the CTF (bipolarity) Group had 22, PSF (schizotypy) Group had 22, and the Control Group had 25 participants.

The three axes of acceleration data collected in this way were then first subjected to a bandpass filtering procedure to filter out low-frequency noise (such as the Earth’s gravity) and high-frequency noise (which might be caused by vibrations, for example). The bandpass filter chosen was the maximally flat magnitude filter or Butterworth filter with cutoff frequencies of 0.25 and 2.5 Hz [46].

The actigraphic devices available on the market use different activity metrics, such as integration-based methods including Activity Count (AC) or Proportional Integration Method (PIM), threshold-based methods such as Zero Crossing Method (ZCM) or Time Above Threshold (TAT), and many others (AI, ENMO, MAD, etc.) [46]. For activity calculation, we have chosen the ZCM metric, which identifies activity based on the number of times a certain threshold is crossed by the acceleration signal. The threshold level should not be necessarily 0 g due to measurement noise; therefore, we have selected a threshold level of 0.05 g which is slightly above the noise level.

An activity metric can be applied to the magnitude of the acceleration or separately to the axial components. However, the resulting activity data produced by the two approaches could differ significantly, in the case of the ZCM metric, this difference is negligible [46]. Therefore, for the ZCM activity calculation, the crossings counted on the three axes were summed into 60 s epochs. To make the ZCM functions smoother, sliding window averaging was applied, using the previous 5 and the subsequent 5 for each epoch. In contrast, for the feature extraction, we used the resultant magnitude of the acceleration signal, which proved to be suitable for the subsequent use of multiple software packages.

### 2.3. Algorithms

When selecting the appropriate categorization algorithms, our team followed the lead of Jakobsen et al. [47], whose 2020 paper attempted to uniquely categorize the two groups based on data collected from schizophrenic and healthy patients. The three categorization algorithms they used, and which we decided to use after initial evaluations, were:Logistic regression (LGR) [48]: one of the most common and simplest algorithms for estimating a categorization function based on discrete values;Random Forest (RF) [49]: categorization is decided by majority voting based on unpruned classification trees grown from random samples of the original data, and choosing a random predictor instead of the best split at each node;Light Gradient Boosting (LGB) [50]: it estimates the categorization function for a given training sample while minimizing the value of a loss function over the joint distribution of all (y,x) pairs of values.

To train and test our samples, we have 10-fold cross-validation [51], in which the sample data are shuffled and divided into 10 groups, each of which is marked as part of either the training or the testing set, with the groups being regenerated for each epoch of the algorithm. The 10-fold cross-validation has shown very good results even on smaller datasets, such as ours.

The next step was to establish a baseline for our experiments with the three algorithms by comparing them individually with the Control Group (C) and the bipolar or schizophrenic group. The three characteristics used by Jakobsen et al. [47], which, according to them, described the subjects in the most general way were the ZCM mean, ZCM standard deviation, and ZCM null-ratio (ZCM data with 0 values). The results can be seen in Table 1. Accuracy is calculated by adding the number of true positives and true negatives and dividing by the total number of outcomes. Precision is calculated by dividing the number of true positives by the total number of true positives and false positives. Recall is calculated by dividing the number of true positives by the total number of true positives and false negatives.

Next, we turned to the detailed workings of these methods to select from the collected actigraphic data the features that could effectively contribute to the correct categorization of groups C, CTF (bipolarity), and PSF (schizotypy). To this end, we first had to determine the method for calculating the importance score. The SHAP or Shapley Additive exPlanations [52], an interpretive method based on Shapley values, was chosen. Shapley was originally used as a proven method in game theory to determine the contribution of each participant to the outcome of a given game. Applied to machine learning as SHAP, it can very effectively determine which of the identified features had the greatest impact on the decision. Theoretically, the Shapley values of a given Machine learning (ML) model are calculated by observing the performance of the model on each feature combination. Based on this, SHAP can calculate the impact of each feature. To omit one (or more) feature(s), SHAP replaces the feature values with random values and then makes a prediction. The impact is calculated by how much the prediction worsens without the given features, not just an impact for the given prediction, but for each sample. In reality, the algorithm does not calculate all possible combinations of features, but only some of them, and then makes an assumption about the overall picture.

Each feature has n Shapley values, where n is the number of decisions made by the model, i.e., the number of samples in the dataset. In our case, one sample corresponds to one individual, so we know the effect of the features in labeling an individual.

Using the Python SHAP library by Lundberg et al. [52], we found that the Shapley values of the three features evolved as shown in Figure 1 using logistic regression, with the Shapley values lying on the x-axis. In this case, the more positive the value, the more the feature (for a given sample) contributed to being classified as disease-prone by the algorithm. Each point represents the effect of a single sample on a single feature, so each feature has n points, where n is the sample number (in this case, the number of contributors). Color-coding is used to show which values of a particular feature (small or large values on its own scale) contributed in which direction and to what extent to the final decision. In this case, susceptibility to bipolarity is indicated by low values for the mean and standard deviation of daily activity. For the detailed SHAP algorithm please refer to Appendix A.

### 2.4. Feature Engineering

Below, we describe the methods used to derive a variety of characteristics from the collected activity data. Because of the comparability of the data and the good applicability of the learning algorithms, we tried several different normalization methods. We tested six different normalization methods on the three algorithms with randomly selected feature combinations, found the standard normalization to be the best, and applied it to the resulting features.

For the first set of features, we used the pyActigraphy library [53], a popular open-source Python package designed specifically for processing actigraphy data. The resulting features were computed using the package’s built-in BaseRaw model functions for the preprocessed magnitude of acceleration data generated from the triaxial data from which the gravitational acceleration bias was removed. These features measure the features most commonly used in the literature to assess vulnerability to mental illness, such as the most active 10 h mean (M10), the least active 5 h mean (L5), and interdaily stability (IS). In total, six such features were developed with the package and they are listed in Table 2.

Next, we performed Savitzky–Golay filtering [54] on these averages, which smoothed the data without distorting the signal trend. After this step, we were able to determine different thresholds from the data and define new features based on them, which one can see in Table 3.

Because we studied individuals without manifest disease, we assumed that the differences between our groups would be much smaller. We tried to find more subtle differences among groups. One way to do this was to characterize the shape of the curves from the ZCM data, which can be seen in Figure 2 and Figure 3. These essentially describe the relief pattern of the motion curve. To avoid excessive outliers, we averaged the ZCM data with a 5-5 long sliding window.

Because these are highly deterministic and sensitive variables, we assumed that diurnal activity would be too variable, so we characterized only the period during sleep with these features. The periods when ZCM values lay between 0 and 5 were considered the longest periods of sleep. Based on this threshold, we obtained realistic distributions of sleep periods. A “peak” in the curve means ZCM data greater than 0, and two peaks are separated by zero values. The peaks were grouped into small and large, different features were calculated, such as the distance between them, lengths, heights. From this, we derived some features, such as minima, maxima, means, and medians, and we separated the small and large peaks in two ways, namely along the median and by the lower and upper quartiles (3 × 4 × 2 = 24 features). The same features were defined for each peak (12 features).

The most common global measure used for characterizing sleep quality is the fragmentation index (“frg_index”). It describes the ratio of mobile time during sleep (cumulative time of the periods of activities higher than the threshold value, plus the time of passive periods less than a minute), as compared to the total sleep time (time from falling asleep until full awakening) [55]. An elevated “frg_index” is found to accompany various mental or physical disorders [55,56], although it does not contain information about the distribution of periods of nocturnal awakening.

For more details on the feature engineering please refer to Appendix A.

To characterize the latter, too, here we introduced an alternative method based on continuous wavelet analysis (CWA). CWA is known to provide both frequency- and time-domain information; hence, it is often used to analyze time series with a strong stochastic nature, such as various biological signals [57], to extract essential information about the dynamic behavior of the biological system. The CWA of a time series applies a so-called “mother wavelet” (MW), a finite oscillation confined in a time window, whose correlation coefficient with the signal is determined at each time point, while MW is shifted along the time axis of the registered signal. MW can be compressed or dilated by a scale factor defining the size of its time window, and the result of CWA is usually represented by a 2D map, where the correlation coefficients (color-coded in Figure 4b) are depicted as a function of running time and the scale factor (corresponding to the width of the time window of MW). Those regions emerging from the map that show enhanced structural features (i.e., high-amplitude correlation coefficients) are indicative of outstanding events at the corresponding time and time window (Figure 4b). In our analysis, the so-called Morlet wavelet was used as the MW, which is essentially a single-frequency sinusoidal inside a Gaussian envelope. The Morlet wavelet, also called the Gabor wavelet, is known to provide the best trade-off between time and frequency resolution [58], and it has been successfully used for the analysis of physiological signals [57,59].

In order to quantitatively characterize nocturnal activity features on the time window scale, we calculated the squared sum of the correlation coefficients provided by the CWA, at each scale factor (spanned from 1 min to 200 min in linear scale) from concatenated sleep periods of 5 subsequent nights (Figure 4c). From these “structure-factor” curves, it is apparent that major features are distinguishable around the 1 h time window; hence, the wavelet-based structure parameters in this study were defined as the integral (“structure_pm”) and the standard deviation (“structure_pm_stdev”) of these curves between the 20 min and 100 min, and the 51 min and 67 min time-window range, respectively.

The numerical calculation of the wavelet transforms was carried out in the Wavelet Toolbox of the MATLAB program [60]. For the detailed description of the wavelet transformations please refer to Appendix A.

### 2.5. Feature Selection

The feature engineering resulted in a total of 96 features, from which we had to select those best suited for the various training tasks in the next step. Since only a few of the numerous features can be used to distinguish groups efficiently, we tried to weed out the less useful features. In our selection, we examined the extent to which the Welch test [61] ranked the expected values of a particular feature as different between the two groups with a tendency toward mental illness and the control group.

The *p*-value (significance level) in our Welch test was 0.3, leaving 46 characteristics out of the 96.

Among these, however, many features were highly correlated. Therefore, we began to create subgroups within which there was no strong correlation and whose individual value in the decision process was therefore also high. To do this, we created the correlation matrix of the 46 features as a weighted edge graph (complete), where the nodes represented the features and the weights of the edges represented the absolute values of the correlation between the two features. We deleted those edges whose weights exceeded a threshold; in our case, it was 0.3.

In the resulting graph, we searched for the complete graphs with the most elements, i.e., clicks with at least 8 elements, resulting in 2513 combinations. We set these two parameters (absolute correlation value and minimum number of nodes) based on the number of subsets created. These were further narrowed down by testing the prediction algorithms with the possible feature combinations (clicks) and creating the final feature set based on their performance. In our case, the computational complexity was solved by randomly selecting 1600 feature combinations. These 1600 feature combinations consist of 45 features. In total, about 9600 models were generated (based on the three algorithms, the two groups, and the 1600 feature combinations), from which we selected those that outperformed the baseline for each algorithm and group combination.

## 3. Results

### 3.1. Machine Learning Results

The results of the models performing above the baseline were collected and their accuracy are plotted on the boxplot in Figure 5.

In the CTF Group (bipolarity), Random Forest showed spectacular improvement over the baseline, with an average increase in accuracy from 5% to a maximum of 17%, a maximum increase of 4% for logistic regression, and a maximum increase of 6.3% for Light Gradient Boost (LGB). The outlier values for the best models are quantified and given in Table 4.

The greatest improvement occurred in the PSF Group (schizotypy), where the maximum accuracy of the three algorithms was increased by an average of 18%, with some exceptions. The best models are listed in Table 5.

The detailed features which were used in these best-performing models are summarized in Table 6 and Table 7.

The correlation table showing the final 46 features is given in Figure 6.

### 3.2. Feature Effects

#### 3.2.1. CTF (Bipolarity Group) Results

In the end, each algorithm (logreg, lgb, rfc) in each group had a few hundred models that performed better than the baseline. This proved difficult to interpret, so we summarized these results. To apply the Shapley summary to a more general model, we summed the Shapley values for the corresponding algorithm, group, and feature row by row. For each feature, each row has a Shapley value, as shown in Figure 1. Thus, when we sum two models, we sum the Shapley values of the same features that belong to the same rows.

This step was necessary because we had a very small dataset. With this method, we can reduce the bias due to random events and the inconsistency of the explanatory power of the models. For example, consider a particular feature, “zero_ratio”, which occurs in two different linear regression models, “model1” and “model2”. An exemplar individual, named “E01”, has a Shapley value with opposite values in the two models. In the first model, the value of the trait indicates the risk of mental illness, but in the second case, it is the opposite. Thus, if we add the first Shapley value (from “model1” of “zero_ratio” of “E01”) and the second (from “model2” of “zero_ratio” of “E01”), they cancel each other out.

Shapley values for the CTF Group are shown in Figure 7. In order of importance, there is a gradual decline in the average importance scores after the first five features. The distribution by feature value is nice, with high values going steadily from high to low values (the coloring is continuous), and values with zero Shapley values are purple, so values in the middle of the scale are not important to the decision. This is just the behavior we would expect from an algorithm such as logistic regression.

With this group, there is a gradual decline in the mean scores for the importance of the Light Gradient Boost feature. Figure 8 shows the key part of the algorithm’s operation, the gradient method of threshold selection (the boundary between presence and absence of susceptibility), in terms of the weighting scores clustered at the edges.

In the case of the random forest depicted in Figure 9, the first few features are also significant, but unlike the previous cases, they are not well separated by feature value, which is to be expected given the small amount of data and the randomly generated decision trees. This is not necessarily far from the potentially plausible solutions (especially since these models performed best), but it could also mean that there is a nonlinear distribution of propensity to bipolarity along our variables. For example, for 3_third, the average activity in the third third of the day, there are very mixed values on both sides of the axis of the graph. This might mean that different values of several variables actually have a low value, a high value, and a slope.

For a summary of Shapley values, see Table 8. For each algorithm, the high intensity of the least active 5 h (L5) was a leading determinant of the propensity to bipolarize. In addition, features describing the characteristics of movements during sleep indicated that movements were relatively infrequent, but when they occurred, they were long-lasting and of high intensity, consistent with the high values of the fragmentation index. Relatively low values were characteristics of the second and third periods of the day (with a higher proportion of zero values). Another significant feature was IS. Here, samples with high IS were much more likely to be classified in the propensity group.

#### 3.2.2. PSF (Schizotypy Group) Results

In the case of logistic regression, the importance of the features decreases more or less linearly, as can be seen in Figure 10, and it can be said that in this algorithm, most of the features had a large impact on the result.

In contrast, it can be seen from Figure 11 that the Light Gradient Boost features are largely separated, with the first zero_ratio being the most important and the next 4–5 features sharing the rest. It is interesting that this algorithm performs particularly well in separating individuals with vulnerability to schizophrenia, and that the zero ratio is given such a high weighting.

Similar to the random forest models in the CTF Group (bipolarity), there is a lot of scatter and overlap in the feature value coloring in Figure 12. The first 4 features are more important, but the following also have significant impact as well.

As for the CTF Group (bipolarity), the Shapley values of the models are summarized in Table 9. In schizophrenia-prone individuals, the corresponding features (L5,2_thrd—low, zero_ratio—high) show that activity is low, but in the third period of the day (the period before sleep, 16:00–00:00), the same value is higher. High values for sleep fragmentation indicate deterioration of sleep quality, but nocturnal features (lower_humps_width_min_qrt+, min_length_between_bigger+, upper_humps_width_max+, upper_humps_(max/min/avg) _distance+—based on these) together with L5 indicate that activity during sleep is low. For structurality (IS), individuals with schizophrenic predisposition show a higher affinity. Part of the results can be explained by the selection of individuals with schizotypy who have negative symptoms.

## 4. Discussion

In the case of the CTF group, the main similarities were the L5, zero_ratio, 3_thrd, and some features of the night movement. However, some features that were considered very important in the Shapley value-based analysis (such as IS, 2_thrd, frg_imdex, and sleep time) proved irrelevant in the Adaptive, Hybrid Feature Selection (AHFS). Just as in the CTF group, there are similarities (zero_ratio, structure_pm, etc.) and differences (IS, 3_thrd, L5, frg_imdex) in the PSF group.

Figure 13 lists the features that were found in the literature for the two mental disorders and are comparable to the features we use in the study.

Most of the variables were found mainly in manifest mental illness or in cases with a very high risk of developing the illness. Our study indicates that some characteristics can be detected even in cases of low susceptibility. We examined daily activity with variables 2_thrd, 3_thrd, and L5 covering approximately 24 h. Our results show a general decrease in daily activity in our two propensity groups, as described in the literature [14,15,27]. Based on the analysis of the data, it is more advisable to process the data in this way, i.e., in diurnal subdivisions, since differences can be found both between the Control Group and the two propensity groups and between the two groups studied in different time periods. However, our models did not find any sufficient explanatory power for the feature describing the average 24 h activity.

The features describing movement during sleep are consistent with fragmentation and give us a better insight into patterns of nocturnal activity (which show significant differences compared with controls). For example, in the CTF Group (latent bipolar liability), we saw that in addition to high values of L5 and the fragmentation index, sleep characteristics were also descriptive: pauses between movements increased, but so did the amplitude and length of each movement. In addition, we noticed that the fragmentation index was also higher, but L5 was low, meaning that the quality of sleep was also worse in this group, but they were more restful during the least active 5 h. The interdaily stability variable also had a similar pattern between groups to that in the literature: schizophrenia-prone individuals showed higher stability, similar to patients diagnosed with schizophrenia studied by Berle et al. [27]; bipolar-prone individuals showed lower stability, similar to patients studied by Jones et al. [28]. Intraday variability and the lower M10 variable [18] were examined but were not among the top 20 features in most models (M10 is the 18th feature in the random forest in the schizophrenia group, but no distribution by feature value is shown).

In order to validate our results, another approach was also used by our team members, called Adaptive, Hybrid Feature Selection or AHFS. This hybrid solution combines the existing (supervised, state-of-the-art) feature selection techniques, which have their own specific but fixed feature evaluation measures/metrics, in an adaptive manner [62]. The AHFS algorithm is a simple method that first selects the “most important” feature, then another, the third, and so on. The difference between the two methods and their results is so large and interesting that this article is not sufficient for a detailed analysis, but we will cover this topic in another article. The AHFS uses artificial neural networks to create a good set of features. This algorithm has significant differences from the other three, so the existence of the differences is not surprising.

## 5. Conclusions

Schizophrenia and bipolar disorder are severe mental disorders whose early detection is not sufficient to halt chronic progression. Another approach is to monitor individuals at risk and their symptoms. However, we wanted to go one step further and group the healthy university student population according to certain characteristics considered to have specificity. Screening and monitoring people can help us better understand the development pathway of these disorders and make it possible to prevent the onset of the first manifest episodes. In summary, the results of our research are as follows:We successfully applied learning algorithms to separate Control, CTF (bipolarity), and PSF (schizotypy) groups;We used features engineered from wavelet analysis that had a significant impact on this decision;Features describing the characteristics of movements during sleep were used and interpreted in the training;Based on a significant number of models, we selected the most efficient ones and then determined the Shapley value of the low-correlating features used in them;The results of the literature were supported and supplemented by new features selected during our teaching.

We successfully applied learning algorithms to separate both the low-risk groups for bipolarity and schizotypy from the control, and in both cases, we achieved an accuracy of about 80%. The biggest limiting factor was the potential for random effects due to the small sample size, which we attempted to filter out using a large number of models. The importance scores of the features in these models were calculated using Shapley values and averaged so that we could predict the weighting and contribution of each feature with a relatively high degree of confidence. Based on these results, we found that the features we constructed have high explanatory power. The nocturnal features may be a good complement for the fragmentation index and L5 and allow us to better understand the higher-resolution activity features for each disease and its liabilities.

## Figures and Tables

**Figure 1 sensors-23-00958-f001:**
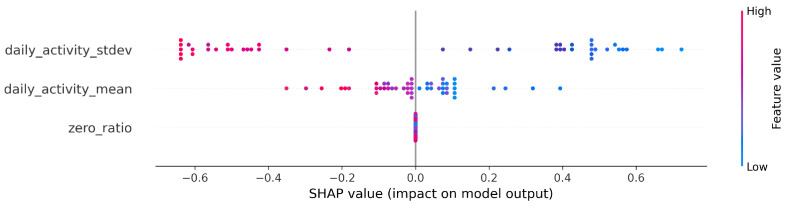
SHAP (Shapley Additive exPlanations) values from the baseline logistic regression.

**Figure 2 sensors-23-00958-f002:**
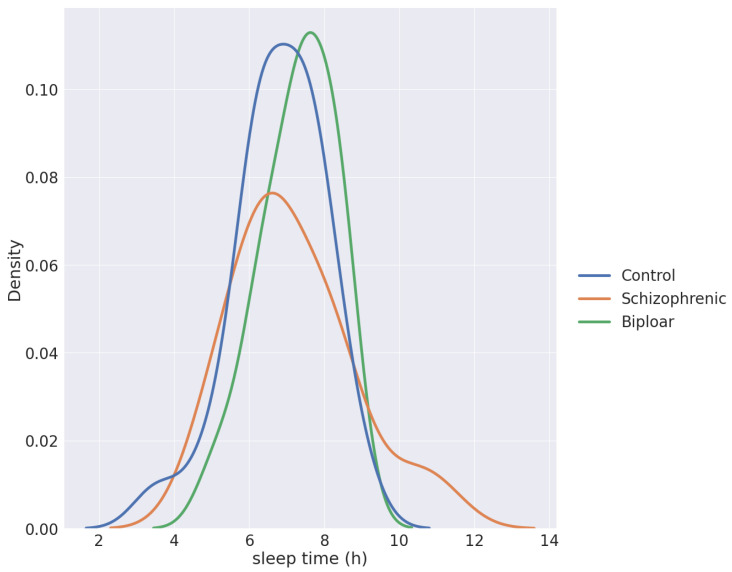
The density function of sleep times per group.

**Figure 3 sensors-23-00958-f003:**
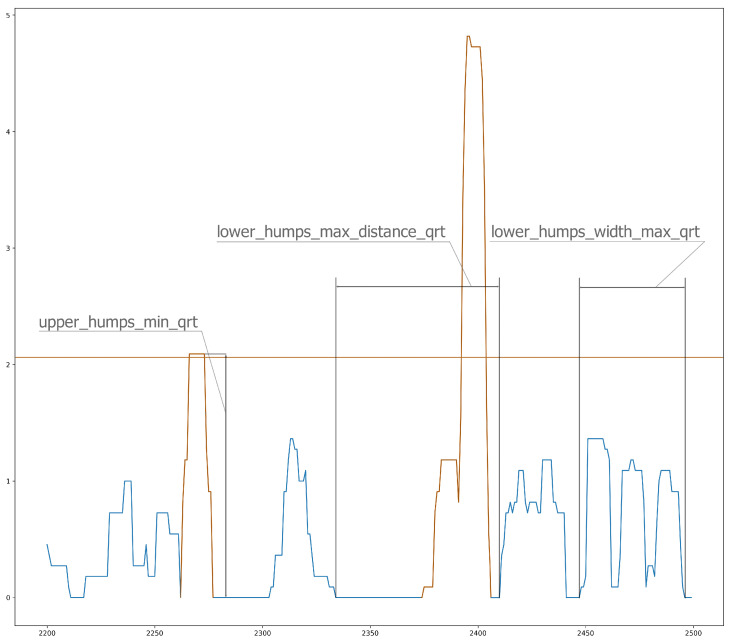
Sliding window analysis of ZCM (Zero Crossing Method) data.

**Figure 4 sensors-23-00958-f004:**
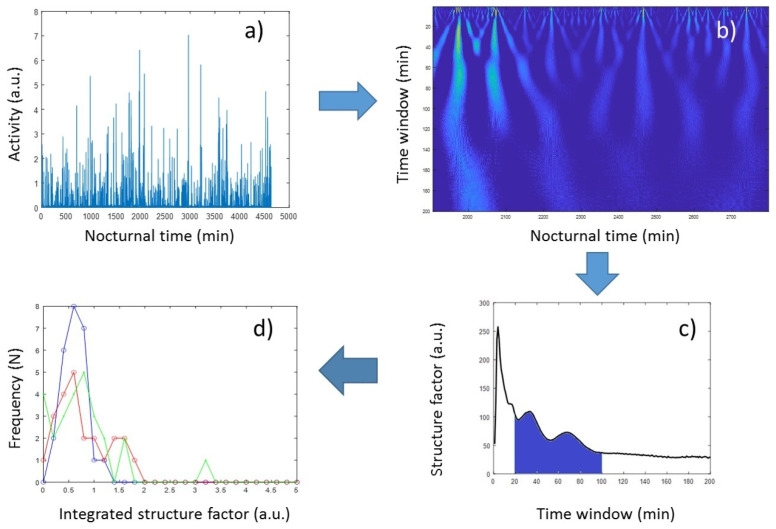
Flowchart of the wavelet-based evaluation of nocturnal activity structures. (**a**) Activities of concatenated sleep periods of 5 subsequent nights. (**b**) Correlation-coefficient map of the time series in (**a**), as a result of continuous wavelet analysis. (**c**) Structure parameters derived from the map in (**b**), as a function of scale parameter in the 1 to 200 s time-window range. (**d**) Distribution of the integrated structure factors (structure_pms) among the three groups of volunteers (Control Group (blue), Cyclothymia Factor Group (red), and Positive Schizotypy Factor Group (green)).

**Figure 5 sensors-23-00958-f005:**
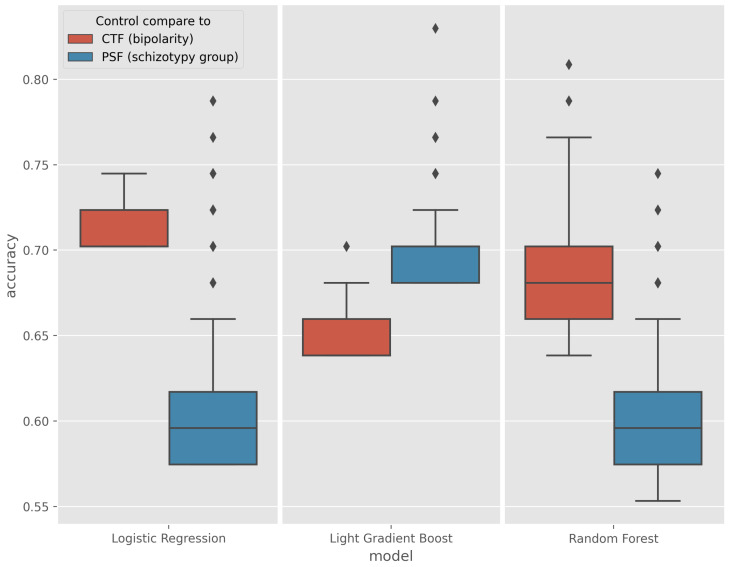
Boxplot of models performing above the baseline in the CTF (Cyclothymia Factor) and PSF (Positive Schizotypy Factor) groups.

**Figure 6 sensors-23-00958-f006:**
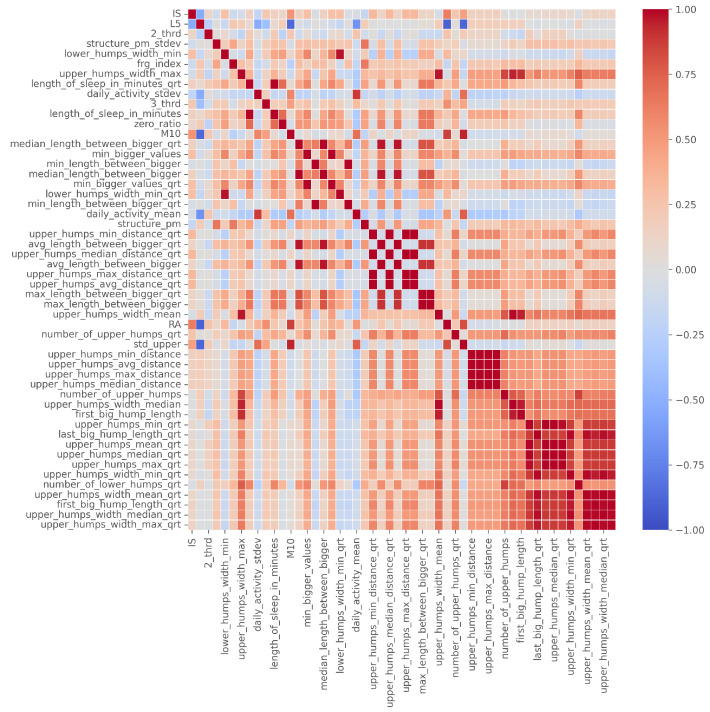
The correlation table of the 46 filtered features, ordered by number of occurrence.

**Figure 7 sensors-23-00958-f007:**
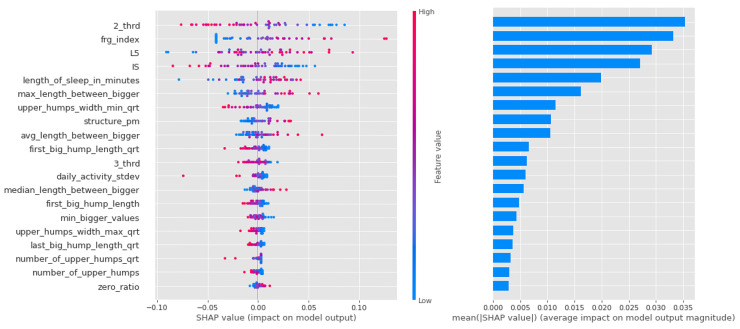
Shapley values of features taken from the best model for Logistic regression in the CTF (Cyclothymia Factor) group.

**Figure 8 sensors-23-00958-f008:**
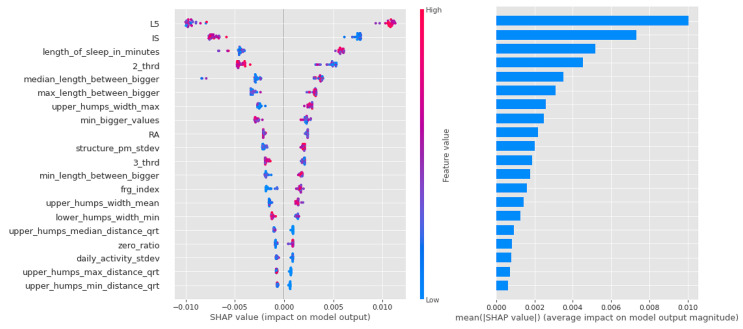
Shapley values of features from the best model for Light Gradient Boost in the CTF (Cyclothymia Factor) group.

**Figure 9 sensors-23-00958-f009:**
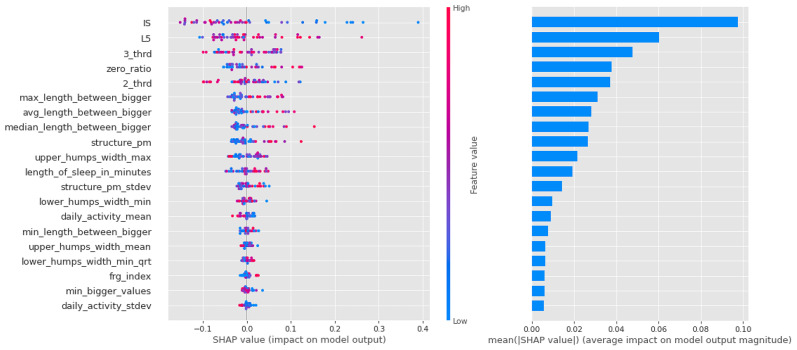
Shapley values of features taken from the best model for Random forest in the CTF (Cyclothymia Factor) group.

**Figure 10 sensors-23-00958-f010:**
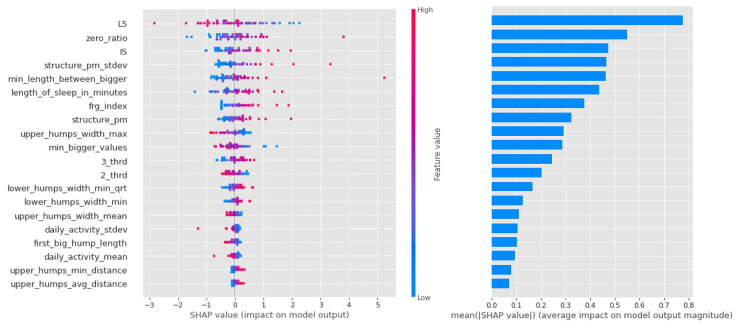
Shapley values of features taken from the best model for Logistic regression in the PSF (Positive Schizotypy Factor) group.

**Figure 11 sensors-23-00958-f011:**
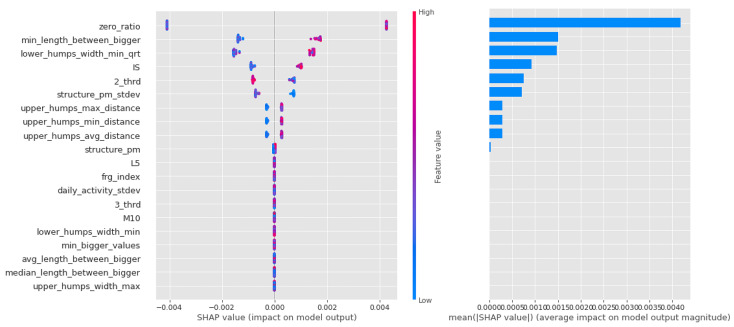
Shapley values of features from the best model for Light Gradient Boost in the PSF (Positive Schizotypy Factor) group.

**Figure 12 sensors-23-00958-f012:**
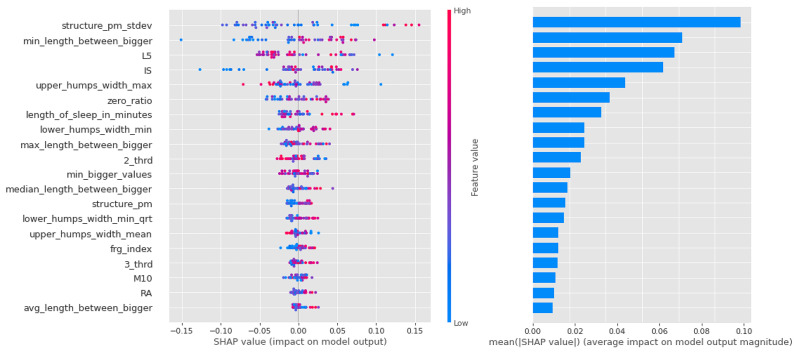
Shapley values of features from the best model for Random forest in the PSF (Positive Schizotypy Factor) group.

**Figure 13 sensors-23-00958-f013:**
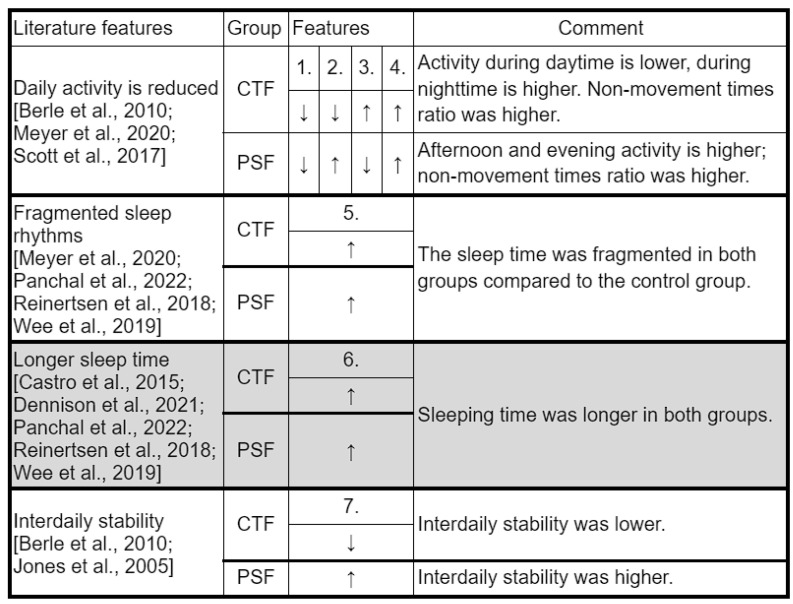
A comparison of our key features with those in the literature. All the features taken from the literature refer only to patients with bipolar or schizophrenic disorder, with the exception of longer sleep duration (highlighted in gray), as this also occurs in at-risk individuals. 1.: 2_thrd;2.: 3_thrd; 3.: L5; 4.: zero_ratio; 5.: frg_index; 6.: length_of sleep_in_minutes; 7.: IS; CTF: Cyclothymia Factor; PSF: Positive Schizotypy Factor. Berle et al., 2010 [27], Castro et al., 2015 [18], Dennison et al., 2021 [22], Meyer et al., 2020 [14], Murray et al., 2020 [23], Panchal et al., 2022 [24], Reinertsen et al., 2018 [25], Scott et al., 2017 [15], Wee et al., 2019 [26], Jones et al., 2005 [28].

**Table 1 sensors-23-00958-t001:** Baseline Results.

Model	Control Compared to	Accuracy	Precision	Recall
Logistic regression	Bipolar	0.680	0.652	0.760
	Schizophrenic	0.531	0.500	0.600
Random forest	Bipolar	0.636	0.660	0.640
	Schizophrenic	0.545	0.638	0.640
Light Gradient Boost	Bipolar	0.681	0.684	0.840
	Schizophrenic	0.745	0.812	0.800

**Table 2 sensors-23-00958-t002:** Features engineered from the pyActigraphy package.

Feature	Description
M10	Daily mean activity of the 10 most active hours
L5	Daily mean activity of the 5 least active hours
RA	Relative rest/activity amplitude.
ADAT	Total average daily activity
IS	Interdaily stability
IV	Intradaily variability

**Table 3 sensors-23-00958-t003:** Features derived following the Savitzky–Golay filtering.

Feature	Description
floor_sum_values	sum of lower boundary values
e_v_avg_value	average boundary value
avg_ceiling_value	average upper boundary value
avg_floor_value	average lower boundary value
avg_ceiling_floor_diff	difference between average upper and average lower boundary values
std_avg	standard deviation of boundary values
std_upper	standard deviation of upper boundary values

**Table 4 sensors-23-00958-t004:** Best-performing models for each algorithm in the CTF (bipolarity) group.

Algorithm	Accuracy	Precision	Recall
Logistic regression	0.744681	0.727273	0.727273
Light Gradient Boost	0.702128	0.900000	0.409091
Random Forest	0.808511	0.809524	0.772727

**Table 5 sensors-23-00958-t005:** Best-performing models for each algorithm in the PSF (schizotypy) group.

Algorithm	Accuracy	Precision	Recall
Logistic regression	0.787234	0.800000	0.727273
Light Gradient Boost	0.829787	0.888889	0.727273
Random Forest	0.744681	0.777778	0.636364

**Table 6 sensors-23-00958-t006:** The list of features used in the best-performing CTF (bipolarity) models, 1 meaning that feature was used in that model and 0 meaning that it was not part of the feature set.

Features	Logistic Regression	Light Gradient Boost	Random Forest
L5	1	1	1
IS	1	1	1
structure_pm_stdev	1	1	1
zero_ratio	1	1	1
min_length_between_bigger	1	0	1
lower_humps_width_min	0	1	0
3_thrd	0	1	0
upper_humps_width_max	0	1	0
median_length_between_bigger	0	1	0
min_bigger_values	0	0	1
2_thrd	0	0	0
upper_humps_median_distance	0	1	0
M10	1	0	0
number_of_upper_humps	0	0	1
upper_humps_median_distance	1	0	0
daily_activity_mean	0	0	1
lower_humps_width_min_qrt	1	0	0

**Table 7 sensors-23-00958-t007:** The list of features used in the best performing PSF (schizophrenic) models, 1 meaning that feature was used in that model and 0 meaning that it was not part of the feature set.

Features	Logistic Regression	Light Gradient Boost	Random Forest
L5	1	1	1
IS	1	1	1
structure_pm_stdev	0	1	1
zero_ratio	0	1	0
min_length_between_bigger	0	1	1
lower_humps_width_min	0	0	1
3_thrd	1	0	0
upper_humps_width_max	0	0	1
median_length_between_bigger	1	0	0
min_bigger_values	0	1	0
2_thrd	1	0	1
upper_humps_median_distance	0	0	1
M10	0	1	0
daily_activity_stdev	0	0	1
max_length_between_bigger	0	0	1
number_of_upper_humps	0	0	0
upper_humps_min_distance	0	1	0
structure_pm	1	0	0
upper_humps_width_max_qrt	1	0	0
length_of_sleep_in_minutes	1	0	0

**Table 8 sensors-23-00958-t008:** Shapley values for best-performing models in the CTF (Cyclothymia Factor) group.

Feature	LGR	LGB	RF
IS	4 −	2 −	1 −
L5	3 +	1 +	2 +
2_thrd	1 −	8 −	5 −
3_thrd	11 −	x ?	4 −
frg_index	2 +		
structure_pm	8 +	x +	7 +
length_of_sleep_in_minutes	5 +	9 +	x +
zero_ratio	x +	3 +	3 +
max_length_between_bigger	6 +	5 +	x +
median_length_between_bigger	x +	6 +	x +
avg_length_between_bigger	9 +		6 +
upper_humps_width_max	x −	4 +	x +
upper_humps_width_min_qrt	7 −		
min_bigger_values		7 −	x −
min_length_bigger		x +	
daily_activity_stdev		x −	x +

LGR: Logistic regression; LGB: Light Gradient Boosting; RF: Random forest.

**Table 9 sensors-23-00958-t009:** Shapley values for best-performing models in the PSF (Positive Schizotypy Factor) group.

Feature	LGR	LGB	RF
L5	1 −		3 −
IS	3 +	4 +	4 +
zero_ratio	2 +	1 +	6 +
frg_index	7 +		x +
structure_pm	8 +		x +
structure_pm_stdev	4 +	6 −	1 +
lower_humps_width_min_qrt	x +	3 +	x +
3_thrd	x +		x +
2_thrd	x −	5 -	x −
min_length_between_bigger	5+	2 +	2 +
daily_activity_stdev	x −		
upper_humps_width_max	x −		5 +
upper_humps_max_distance		7+	
upper_humps_min_distance		8+	
upper_humps_avg_distance		9+	
length_of_sleep_in_minutes	6+		7 +

LGR: Logistic regression; LGB: Light Gradient Boosting; RF: Random forest.

## Data Availability

Our analysis presented in this study is based on the measurement data, collected by Maczák et al. [46], which are publicly available. The 42 10-day-long raw triaxial acceleration signals (measured on different healthy human subjects’ non-dominant wrists, sampled at 10 Hz in the ±8 g measurement interval) are downloadable from Figshare under CC-BY 4.0 license through the following DOI: 10.6084/m9.figshare.16437684.

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
