# Peer review of "The Actigraphy-Based Identification of Premorbid Latent Liability of Schizophrenia and Bipolar Disorder"

_sensors, 2023, doi:10.3390/s23020958_

Round 1
Reviewer 1 Report
This study developed a small wrist-worn measurement device based on an accelerometer that collects and identifies sensor signals and, through a machine learning model, is able to detect premorbid potential risks for schizophrenia and bipolar disorder.
Author Response
Dear Reviewer,
Thank you for your positive review.
Sincerely,
Ádám Nagy
Reviewer 2 Report
Thank you very much for opportunity to read this interesting paper entitled : “Actigraphy based Identification of Premorbid Latent Liability of Schizophrenia and Bipolar Disorder” by Nagy et al.
This is very interesting paper, well written and in scientific soundness.
Introduction is filled with sufficient information about the topic. Review of the literature is done properly. Material and methods are pretty clear to me, and conclussions arise from the results. High-tech devices are most frequently used in different fields of medicine.
I have minor comments :
1) Please spell out all of abbreviations in the legends below the tables and figures, like CTF etc.
2) I would strongly recommend to extend reference section for this publication : https://doi.org/10.5114/ada.2022.119970 in the introduction section. This will confirm that the use of modern methods has an impact on the mental state of patients in various medical specialities, which will further emphasise your work.
All in all, paper is very good, and should be accepted after minor revision.
Author Response
Dear Reviewer,
Thank you for your positive review.
In accordance with your recommendations, we spelled out all of the abbreviations in the legends below the tables and figures. We also added an abbreviation section to help readers understand them. The suggested publication is very exciting, but in our opinion, it is significantly outside the focus of our manuscript. Since we also deal with eye-tracking research, we will take the proposed article into account when publishing them.
Thank you again for the constructive comments, they helped us to improve our work.
Sincerely,
Ádám Nagy
Reviewer 3 Report
The paper entitled "Actigraphy based Identification of Premorbid Latent Liability of Schizophrenia and Bipolar Disorder" by Nagy et al. appears very well written and conducted with the appropriate scientific principles. Even if the idea is not original, but it brings back all the previous works present in the literature on the subject. The results are very well represented and comprehensive. The discussion accompanied by numerous bibliographic references was well conducted.
Author Response

(The authors gave the same response as above.)

Reviewer 4 Report
The studies described in the manuscript "Actigraphy based Identification of Premorbid Latent Liability of Schizophrenia and Bipolar Disorder" by Nagy et al. examined actigraphic measures to identify features that can be extracted from them so that a machine learning model can detect premorbid latent liabilities for schizotypy and bipolarity. This manuscript is well-written, and the results are explained properly. The followings are some suggestions/concerns:
1. Please explain the term “actigraphy” before start using it in the manuscript.
2. The authors may consider citing the following references-
Eur Psychiatry. 2022 Jun; 65(Suppl 1): S683.
Schizophr Bull. 2018 Feb 15;44(2):286-296.
3. Out of 182 subjects recruited in the study what was the number of males and females?
Author Response
Dear Reviewer,
Thank you for your positive review.
In accordance your recommendation, we explained the term “actigraphy” in the manuscript. To be even more clear we modified the two paragraphs from line 294 to 309, and a sentence from line 376 to 370.
Both proposed research came to exciting insights. In our study, we examined non-help-seeking university students who did not meet illness or clinical risk status (attenuated psychosis) criteria, based on assumed premorbid, latent predisposition traits. Both publications suggested by the reviewer examined people seeking help who were not premorbid, nor prodromal, but already beyond that stage, with CHR status prior to the transition to psychosis. The study presented in the conference poster looked for high clinical risk and its possible premorbid correlations among sick, hospitalized depressed young people. Very good target recognition, and a very good research focus, we will wait for the journal publication as well, and we will follow their work in our research. The second publication, the PREDICT study, also examines help-seekers in a high clinical risk state and analyzes the correlations of psychotic transitions. In our opinion, the results of both articles are far from the focus of our study. At the same time, we consider both relevant, and since we also conduct prodromal research and interventions, we take the proposed papers into consideration when publishing our later findings.
Finally we did not analyze the data of the N=182 persons selected based on the inclusion criteria, only those persons who were enrolled in the study were examined. More than half of those who met the inclusion criteria immediately dropped out based on the exclusion criteria, and we did not conduct an analysis of their data.
Thank you again for the constructive comments, they helped us to improve our work.
Sincerely,
Ádám Nagy